# Maturity Assessment of Intelligent Construction Management

**Chao Lin** [1], **Zhen-Zhong Hu** [1], **Cheng Yang** [2], **Yi-Chuan Deng** [3], **Wei Zheng** [2] and **Jia-Rui Lin** [4,*]

1   Tsinghua Shenzhen International Graduate School, Tsinghua University, Shenzhen 518055, China
2   Zhejiang Supervision on Highway and Water Transportation Construction Engineering Co., Ltd., Hangzhou 310000, China
3   School of Civil Engineering and Transportation, South China University of Technology, Guangzhou 510641, China
4   Department of Civil Engineering, Tsinghua University, Beijing 100084, China
*   Correspondence: lin611@tsinghua.edu.cn

**Abstract:** In the new era of Construction 4.0, the application of a large number of intelligent information technologies (ITs) and advanced managerial approaches have brought about the rapid development of intelligent construction management (ICM). However, it is still unclear how to assess the maturity of ICM. In this study, a maturity assessment system for ICM was formulated through literature reviews, questionnaires, expert discussions and a case study. A maturity scoring table containing five assessment dimensions and twenty assessment indicators was developed, and corresponding maturity levels and a radar chart of dimensions were designed. A case study of the assessments of two construction enterprises was conducted to validate that the proposed assessment system could be used by construction enterprises to quantitatively assess their ICM maturities and obtain both overall and specific assessment results. This study also proposed practical improvement methods to improve ICM maturities for construction enterprises with different maturity levels. Furthermore, the study also discussed the development direction of ICM at present and in the short-term future, which should be paid more attention to by the construction industry.

**Keywords:** intelligent construction management (ICM); construction industry; maturity assessment system; improvement plan





## 1. Introduction

The construction industry is a traditional pillar industry in many countries, and its contribution to economic growth and long-term national development is widely acknowledged [1]. In China, for example, the construction industry contributed about 25.6% to the national gross domestic product (GDP) in 2021 [2]. However, the construction industry involves a large number of participants and covers multiple professions, so improper and bad management of any aspect often causes huge losses. The fatal injury rate for the construction industry is higher than the average for all other industries due to its labor-intensive characteristics and poor safety management during production processes [3]. Careless quality management will cause a hidden danger to the later operation of structures. Many construction projects worldwide were completed with significant time and cost overruns due to bad schedule management [4]. Furthermore, conflicts, disputes and arbitration between construction parties caused by poor construction management greatly lower the construction productivity on site. On the contrary, proper construction management can reduce potential risks when implementing investment and construction projects and make necessary conditions for the timely and high-quality delivery of projects within the planned budget. Construction management is a professional service that provides a project's owner(s) with effective management of the project's schedule, cost, quality, safety, scope and function [5], and it plays an increasingly important role in various construction projects.

With the rise of Industry 4.0 [6], the rapid development of information technologies (ITs) has greatly promoted and improved the construction industry. As a result, the terms Construction 4.0 and intelligent construction came into being. Construction 4.0 is a concept that was proposed in reference to Industry 4.0. The definition of Construction 4.0 is dynamically evolving. For example, Sawhney et al. [7] defined Construction 4.0 as a framework that is a confluence and convergence of three broad themes: industrial production, cyber-physical systems and digital and computing technologies. Wu et al. [8] regarded Construction 4.0 as the integration of information and automation technologies in construction projects. There are many advanced technologies involved in Construction 4.0, and Forcael et al. [9] concluded that four essential technologies are needed to understand Construction 4.0 at the present time: 3D printing, big data, virtual reality (VR) and the Internet of Things (IoT). Except for advanced technologies, Construction 4.0 also brought advanced managerial approaches; García de Soto et al. [10] indicated that Construction 4.0 pushes construction organizations and roles to be transformed in many aspects. The evolution from digitalization to intelligence is the mainstream of the development of Industry 4.0 [11]. As a derivative of Industry 4.0, the development direction of Construction 4.0 is the same, so intelligent construction is the ultimate goal of Construction 4.0. The comprehensive development of intelligent construction requires intelligence in every segment, among which intelligent construction management (ICM) plays an essential and inevitable part; it is the foundation of Construction 4.0 and intelligent construction. ICM is the intelligent pattern of construction management; it is a comprehensive evolution of traditional construction management in management concepts, working mode and supporting measures which are achieved by the introduction of intelligent ITs and congenial managerial approaches.

Maturity is the competency, capability and level of sophistication of a selected domain based on a comprehensive set of criteria [12]. The ICM maturity of a certain construction enterprise is its ability to conduct ICM, and it should be considered comprehensively from the technological perspective and from the managerial perspective. Therefore, the maturity assessment of ICM is the comprehensive consideration of the development condition of IT and the application condition of managerial approaches. The purpose of maturity assessment is to identify a gap that can then be closed by subsequent improvement actions [13]. Many construction enterprises have been developing ICM, and the fierce competition among them requires more efficient improvement plans for their ICM maturities. Only when the ICM maturity is accurately assessed can an enterprise select IT and managerial approaches it needs to improve rather than extensively and aimlessly involving all kinds of intelligent ITs and managerial approaches, leading to a waste of human, material and financial resources. Therefore, the maturity assessment of ICM is of great significance for construction enterprises to find out shortcomings and formulate future improvement plans thereafter.

However, there is still a lack of effective systems, methods or even indicators to systematically assess the maturity of ICM, which has encumbered the development of the construction industry. Existing studies and explorations towards ICM just focused on the innovation or application of one or several types of IT. Due to the differences between construction enterprises or projects, as well as the diversity and complexity of advanced ITs, the application depth and breadth of relevant IT are different, and their values and benefits remain uncertain. At the same time, the introduction of advanced IT often leads to a change in managerial approach, including organizational form and workflow. The mismatch between the managerial approach and IT may also greatly limit the efficiency and value of ICM. Therefore, it is difficult to effectively assess the ICM maturities between different construction enterprises and discover their potential problems at the same time. In contrast, available maturity assessment models are increasingly being applied in other informational, digital or intelligent fields as approaches for continuous process improvements [14], such as the building information modeling (BIM) capability maturity model [15] and the digital maturity model [16]. These maturity models enable relevant organizations to clearly understand their development maturity and to make appropriate developing plans later.

In view of the above problems, this study formulated a maturity assessment system for ICM. An intelligent maturity scoring table was established for the quantitative maturity assessment of ICM. The scoring table consisted of five assessment dimensions and twenty assessment indicators. To present the assessment results in both overall and specific aspects, the levels of ICM maturity were set, and the radar chart of assessment dimensions was designed. Finally, a case study of the assessments of two construction enterprises was conducted to validate the usage of the proposed assessment system and intelligent maturity improvement strategies were discussed. The assessment system can be used for leaders in construction enterprises to assess their ICM maturities and obtain vivid assessment results as well as improvement plans. For every construction enterprise, the scoring table transformed its ICM maturity into a score. The corresponding maturity level plotted the position of its ICM maturity in the whole industry. The radar chart of dimensions visualized its strengths and weaknesses in dimensions. Finally, the improvement strategies guided it to improve its ICM maturity according to the assessment results.

The rest of this paper is organized as follows. Section 2 reviews and summarizes previous studies related to ICM and mature assessment systems in other fields. Section 3 introduces the methodology of this research to formulate the maturity assessment system. Section 4 presents the rationality and effectiveness of the proposed maturity assessment system through expert discussion. Section 5 enumerates the components of this maturity assessment system. Section 6 recounts a case study to validate the usage of the proposed assessment systems and discusses the methods to improve ICM maturity. Finally, Section 7 summarizes the main contributions, limitations and future improvements of this research.

## 2. Literature Review

In this section, studies concerning ICM are reviewed, and so are investigations about assessment systems, including methods and models in other informational, digital or intelligent fields to show mature examples.

### 2.1. ICM

Wu et al. [8] emphasized that Construction 4.0 heavily relies on data to build and maintain the interaction between the physical and virtual worlds. Because intelligent construction is the ultimate goal of Construction 4.0, data is also essential for ICM [17]. Intelligent IT for data collection, transmission, aggregation, analysis and sharing can contribute to ICM [18], so can advanced managerial approaches supporting the data-oriented work mode, such as corresponding working post setting and personnel training, online personnel management and workflow interaction, etc. Therefore, the essence of ICM can be concluded as the review and feedback of various types of relevant construction information and data, which includes the collection, transmission and statistics of them, with the support of visualization, intelligent analysis and other technical means in this process.

A number of researchers have investigated the attributes and development direction of ICM from the perspectives of intelligent IT and advanced managerial approaches, respectively. Aiming at intelligent IT, Sawhney et al. [7] illustrated representative IT that is used in Construction 4.0: BIM, cloud-based project management, augmented reality (AR), VR, artificial intelligence (AI), cybersecurity, big data and analytics, blockchain, laser scanner, IoT, etc. These ITs can also be applied to ICM. Aiming at managerial approaches, Woo et al. [19] reviewed different construction management methods by analyzing the efficiency of various methods currently applied to public construction projects. They concluded that the direct supervision method is the most efficient construction management method because of lower cost and less time. García de Soto et al. [10] analyzed the transformation of construction organizations and roles in Construction 4.0. Existing roles evolved, and new roles were created; for example, more employees with digital skills were needed. Many kinds of traditional construction work were automated with the application of robotic systems. Furthermore, current fragmented projects evolved into project-based integrations and eventually into a platform-based integration.

There are other studies that focused on the application of just one certain IT or managerial approach towards ICM. In this study, we reviewed nearly all the existing IT or managerial approaches from the literature. Furthermore, we also discussed with experts in the construction industry to supplement novel IT or managerial approaches which have not been published yet. All ITs or managerial approaches researched are presented in Table 1. Their effects on ICM and sources are also listed.

**Table 1.** IT or managerial approaches supporting ICM.

| Effect on ICM | IT or Managerial Approach | Source |
|---|---|---|
| Management platform | Platforms with terminals for a personal computer (PC), mobile and website | [20,21] |
| | Use a firewall and virus scan against intrusion | [22] |
| | AI voice assistant | [23] |
| | Application of 5G technology | [24] |
| Personnel management | Intelligent attendance system | [25] |
| | Human resource training and assessment | [26,27] |
| | Manage personnel information and user permissions in the platform | [28] |
| | Monitoring of personnel health and performance | [29,30] |
| | Warning of overdue personnel age and qualification | Expert Discussion |
| | Incorporation of COVID-19 guidelines into site health policies | [31] |
| Visualization | Establish BIM or digital twin (DT) in the platform | [20] |
| | Construction simulation in a multidimensional BIM environment | [32] |
| | Construction information sharing in the platform | [33] |
| | Application of VR, AR and mixed reality (MR) | [34,35] |
| | Information carrier and displayed on the site | [36] |
| Workflow | Submit and receive information through the platform | [37] |
| | Fill and modify documents in the platform | [38] |
| | Task management through the platform | [36] |
| | High-performance communication facilities on site | [39] |
| Production | Machinery operation and work tracking and monitoring | [40,41] |
| | Materials management using emerging technologies | [42] |
| Environmental impact | Waste and pollutant monitoring on site | [43,44] |
| | Site workplace environmental situation monitoring | [45] |
| Quality control | Automated data acquisition technologies on the site | [46] |
| | Application of robots | [47] |
| | Mark locations of quality problems in the models | [36] |
| | Declared quality problems tracking | [48] |
| | Vision-based inspection and real-time quality assessment | [49,50] |
| | Application of personal mobile devices | [51] |
| Schedule and contract | Real-time schedule, contract and payment tracking and monitoring | [52,53] |
| | Warning of overdue schedule and contract | [54] |
| Time and cost | Optimization of time and cost using a learning curve | [55] |
| Information management | Record of engineering data and personnel operation | [56] |
| | Information decentralization, forgery and alteration prevention | [57] |
| | Intelligent search engine | [58] |
| | Data integration and simplification | [59] |
| | Application of information extraction (IE) | [60] |
| Work safety | Real-time video surveillance on the site | [61,62] |
| | Worker safety device makes warning in proximity to certain areas | [63] |
| | Warning of unsafe behavior by real-time smart video surveillance | [61,62] |
| | Equipment collision prevention | [64] |
| | Warning of real-time fire, smoke, etc., on the site | [65] |
| | Warning of abnormal value in data collected | [66] |
| Construction coordination | Dispatch list by intelligent work breakdown structure (WBS) calculation | Expert Discussion |
| | Time-space conflicts management | [67] |
| Risk prevention | Preventive measures with the use of the prediction model | [68] |

While all existing studies individually render positive influences on ICM, their research directions, in general, are too scattered to establish sufficient cooperation and connection with each other. Specifically, they suffer from several shortcomings—they

- Neglect the combined effects of IT and managerial approach;
- Do not summarize all ITs and managerial approaches available for ICM;
- Lack methods to assess the application maturities of IT and managerial approaches in ICM;
- Lack practical and appropriate plans to improve the ICM maturity of construction enterprises.

In this study, existing intelligent IT and advanced managerial approaches available for ICM were reviewed, and the combined effect of IT and managerial approaches were considered. Therefore, the assessment system could be established by extracting assessment objects from these contents, and then maturity improvement plans were provided.

### 2.2. Assessment Systems in Other Fields

This study reviewed some representative maturity assessment systems in other informational, digital or intelligent fields, as listed in Table 2. Berghaus and Back [69] indicated that a maturity assessment model should consist of dimensions and criteria that describe the areas of action and maturity stages that indicate the evolution path toward maturity. Though these assessment systems have different assessment targets using different assessment methods, they all set assessment dimensions and criteria, as well as corresponding development guides, to improve maturity. Furthermore, [15,70] set maturity levels to present the overall assessment results and [70] designed a radar chart of dimensions to visualize the strengths and weaknesses in each dimension. The advantage and priority of each assessment system can offer important references to the assessment methods needed in this research: (1) Scoring on dimensions is a quantitative assessment method that has been proven popular and easy to use. (2) Dimension(s) to assess the management capacity should be set, including organizational framework, personnel management, workflow, etc. (3) Assessment results should be displayed clearly from both overall and specific perspectives. For example, maturity levels and radar charts of dimensions can be applied for the overall and specific perspective.

**Table 2.** Maturity assessment systems in other informational, digital or intelligent fields.

| Research | Assessment Target | Assessment Method | Dimension | Advantage and Priority |
|---|---|---|---|---|
| [71,72] | BIM adoption across markets | Score on dimensions | 5 | Comprehensive consideration of policies, management and technologies |
| [15] | BIM capability maturity | Score on dimensions | 11 | Needed dimensions can be selected from the given 11 dimensions |
| [16] | Digital maturity for companies | Single choice questions | 2 | Rapid assessment process |
| [73] | Digital readiness maturity for manufacturing | Yes or no questions | 5 | High objectivity |
| [74] | Project complexity | Analyze from dimensions | 5 | Detailed assessment results |
| [75] | Digital maturity of construction projects | Score on dimensions | 4 | Introduction of the frequency of assessment objects |
| [70] | Digital maturity on construction site | Score on dimensions | 11 | Comprehensive assessment objects |

### 3. Methodology

In this study, assessment indicators were set with criteria from different dimensions. The steps to develop the assessment system are listed below in Figure 1.

The first step was the determination of assessment indicators. In the whole assessment system, the maturity scoring table was the most important part, whose basic elements were assessment indicators. There were a large number of assessment indicators extracted, so it was necessary to set up assessment dimensions and reasonably classify all indicators to facilitate the viewing and use of them and also help construction enterprises to assess their own ICM maturity from the perspective of each dimension.

The second step was the calculation of the weights of assessment indicators. This study used questionnaires designed in correspondence with the precedence chart method

(PCM) [76,77] to consult experts on each indicator's importance to the maturity assessment of ICM, and then the weight of each assessment indicator and dimension could be calculated with the results of the questionnaires. Scores of the assessment indicators and dimensions in the maturity scoring table could later be calculated by converting their weights.

The third step was the design of the maturity scoring table. The indicators could not be directly used, so the assessment criteria were set to instruct assessors when scoring. Arranging each indicator according to its dimension and then adding the corresponding score and assessment criterion made a complete maturity scoring table. Necessary instructions for each content should also be written to guide assessors to use it correctly.

The last step was the analysis of the assessment results. The presentation of assessment results should take into account overall and specific aspects. This research used maturity levels to plot the position of the ICM maturity in the whole industry and a radar chart [78] of dimensions to visualize the strengths and weaknesses in each dimension. Therefore, this step included the setting of appropriate maturity levels and the corresponding relationship between score intervals and maturity levels, as well as the design of the radar chart.

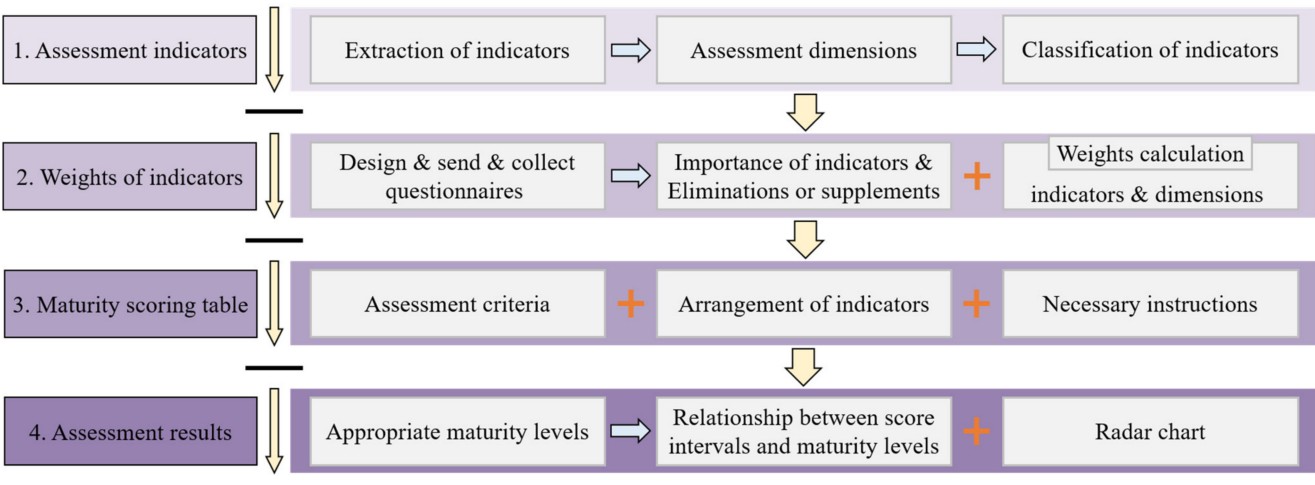

**Figure 1.** The construction process of the assessment system.

### 3.1. Determination of the Assessment Indicators

A complete assessment indicator included the assessment object, weight and criterion. The scope of construction management was confined, but intelligent ITs and advanced managerial approaches supporting ICM over the scope at present were unlimited and uncountable, not to mention new ones are being developed. The responsibility of indicators was to screen objects and contents that could best reflect the development of ICM. Therefore, assessment indicators in this study did not include ITs and managerial approaches available for the scope but abstract attributes that reflect the developing situation of these ITs and managerial approaches instead. In a word, it is not detailed ITs, and managerial approaches that enterprises use but attributes enterprises satisfy that develop ICM.

#### 3.1.1. Extraction of Assessment Indicators

The extraction of assessment indicators needed to comprehensively contain factors from the following three aspects.

(1) Construction management scope to determine the assessment scope of indicators. According to the regulations and requirements of the construction industry, aspects and fields that construction management should be responsible for are clarified, which should be covered by assessment indicators.

(2) ITs and managerial approaches supported ICM to abstract attributes as assessment objects. Comprehension of the application status of relevant IT and managerial

approaches could fully tap the application potential and highest maturity of each one, that is, the scale of attributes.

(3)  Existing assessment methods and systems refer to successful experiences. As mentioned, there were already advanced maturity assessment methods and systems in other fields, as shown in Table 2. Among them were successful experiences in indicator extraction, assessment dimension setting and assessment methods.

The above factors were extracted from both literature and expert discussions. Referring to the literature provided a comprehensive grasp of the relevant contents, and discussing with experts supplemented details omitted in the literature. The latest management technologies are not published yet, and these matters require attention for practical application. These factors should be considered together during collection. First, discover ITs and managerial approaches that could be applied to fields according to the management scope, as shown in Table 1. Then search the application for relevant ITs and managerial approaches to discern their abstract attributes, which are regarded as preliminary assessment objects. Finally, with reference to the framework of other maturity assessment systems, establish indicators by adjusting and reorganizing preliminary assessment objects according to service objects of all IT and managerial approaches. Ensure that the complete independence of indicators was obtained and there was no overlap between them. The indicator extraction process is shown in Figure 2. The assessment indicators were extracted from the literature and expert discussions through these three procedures. Each indicator does not have direct sources of literature or expert discussion because of these fused procedures.

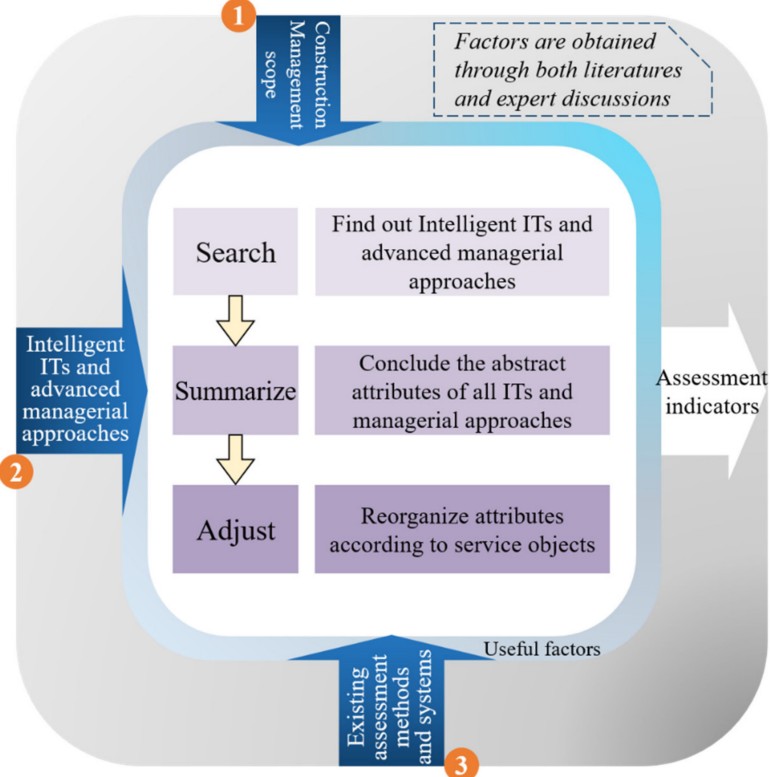

**Figure 2.** The indicator extraction process.

### 3.1.2. Setting of Assessment Dimension and Indicator Classification

The literature review indicated that a dimension to assess the management capacity should be set. Among all indicators, there were some that described the personnel organization and management or workflow of construction management, so first, we set a dimension for them. The remaining indicators described ITs and managerial approaches related to construction management itself. Blanco et al. [79] illustrated many specific and

clear activities to different technologies used during different construction phases, including the phases of design, preconstruction, construction and operations. These specific activities classified ITs and managerial approaches for construction management properly, but they were too scattered. According to the essence of ICM, dimensions for the remaining indicators could be set by composing these specific activities (definitions of these specific activities can be seen in [79]). Considering the service objects and application fields of the remaining indicators, set four dimensions for them. Each dimension and its components are shown in Table 3. Five assessment dimensions were set following strict internal logic to ensure that there was no overlap between each other. The meaning and description of each assessment dimension are shown in Table 4.

**Table 3.** Assessment dimensions and their components for IT and managerial approaches.

| Assessment Dimension | Activities in [79] |
| --- | --- |
| Information collection and monitoring | Materials management, equipment management |
| Information transmission and aggregation | Field productivity, performance dashboard |
| Decision-making supported by visualization | Digital design, design management, contract management, document management |
| Intelligent analysis and deduction | Estimating, construction relationship management, market intelligence, quality control, safety |

**Table 4.** Meaning and description of each assessment dimension.

| Assessment Dimension | Meaning and Description |
| --- | --- |
| I. Organizational framework and working process | A more suitable organizational framework, more powerful personnel management and more efficient work mode are required by ICM. |
| II. Information collection and monitoring | Collection and monitoring of various types of construction information and data through collection and measurement equipment arranged on the construction site. |
| III. Information transmission and aggregation | Transmission and aggregation of information and data collected on-site within the time limit, proper storage and archiving to prevent loss and tampering. |
| IV. Decision-making supported by visualization | Visualization and modeling of engineering information and simulation of construction operation to support decision-making. |
| V. Intelligent analysis and deduction | Analyze engineering information with the application of intelligent technologies to provide calculation, detection, prediction, optimization, etc. |

In order to make each indicator more consistent with the meaning and description of the corresponding dimension when classifying, indicators were appropriately adjusted after determining the dimensions so that each indicator was clearly and uniquely classified into a certain dimension, and the number of indicators contained in each dimension was as close as possible. After the determination of dimensions and the adjustment of indicators, available indicators can be classified into each dimension.

### 3.2. Calculation of the Weights of Assessment Indicators

The questionnaire in this study mainly investigated respondents from three aspects: (1) Basic information of respondents, including professional field and title, working post and year, to show the objectivity of the questionnaire. (2) Eliminations or supplements for existing indicators and judgment of the suitability of the setting of the assessment dimensions and the classification of each assessment indicator to ensure the rationality of the assessment dimensions and indicators. (3) The consultation of respondents about the importance of each indicator to assess the maturity of the ICM. The questionnaire is designed following the PCM. A seven-point Likert scale [80] was used for respondents to choose from very low to very high on the importance of each assessment indicator, where each choice corresponded to a point from one to seven, as shown in Figure 3.

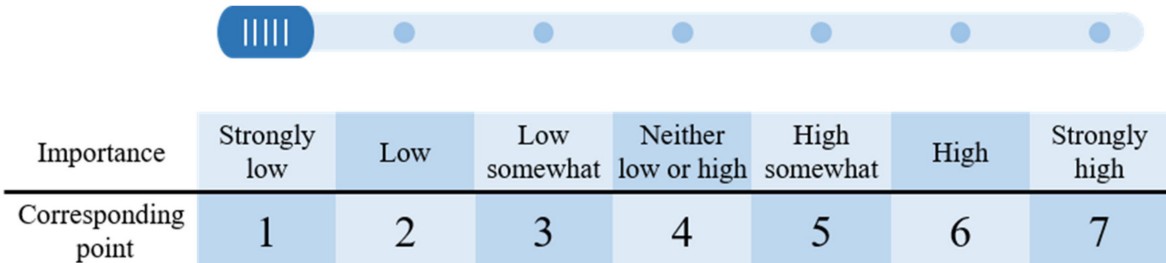

**Figure 3.** The seven-point Likert scale.

The PCM used a comparison matrix to calculate the weights of target objects, as shown in Table 5. The left columns of the table refer to the comparing objects, while the upper rows are the compared objects. In this study, the respondents' opinions on the importance of each assessment indicator were converted into a point from one to seven by the seven-point Likert scale. If there are $n$ indicators for comparison and the average points of all respondents were calculated, it is easy to know that the indicator with the higher point possesses higher importance. Choose $I_1$ and $I_4$ as an example for a pairwise comparison: if $I_1$ is more important, then $a_{14} = 1$ and $a_{41} = 0$; if $I_4$ is more important, then $a_{14} = 0$ and $a_{41} = 1$; if $I_1$ and $I_4$ are equally important, then $a_{14} = a_{41} = 0.5$. Finally, the weight of each indicator can be calculated:

$$w_i = \frac{s_i}{\sum_{i=1}^{n} s_i}$$

In this function, $w_i$ is the weight of the indicator $i$, $s_i$ is the sum of all elements in the row $i$.

**Table 5.** Comparison matrix of PCM.

| Comparison Indicator | $I_1$ | $I_2$ | $I_3$ | $I_4$ | $\ldots$ | $I_n$ | Sum |
|---|---|---|---|---|---|---|---|
| $I_1$ | $a_{11} = 0.5$ | $a_{12}$ | $a_{13}$ | $a_{14}$ | | $a_{1n}$ | $s_1 = \sum_{i=1}^{n} a_{1i}$ |
| $I_2$ | | 0.5 | | | | | |
| $I_3$ | | | 0.5 | | | | |
| $I_4$ | $a_{41}$ | | | 0.5 | | | |
| $\ldots$ | | | | | 0.5 | | |
| $I_n$ | | | | | | 0.5 | |

### 3.3. Design of the Scoring Table

The assessment indicator itself was the summary of ITs and managerial approaches that construction enterprises used, and it does not contain the description of the highest intelligent maturity of each IT or managerial approach. Setting assessment criteria for each indicator was essential for assessors to make more accurate judgments when scoring each assessment indicator. After that, the assessment indicators with scores and assessment criteria could be reasonably arranged according to their dimensions. Finally, the complete scoring table was finished when the necessary instructions for each content were written for correct use.

### 3.4. Analysis of Assessment Results

The scores obtained by the assessors on the assessment of construction enterprises using the scoring table represent their ICM maturities. The score intuitively reflected the overall ICM maturity of each construction enterprise; however, as each enterprise commonly only knows its own score, it cannot plot its position in the whole industry without comparison with other enterprises. Besides, its strengths and weaknesses between different assessment dimensions remain unclarified.

In this study, scores were converted into corresponding maturity levels as the overall presentation of assessment results. The division of maturity levels must ensure that

enterprises at the same level possess ICM maturities at roughly the same standard. How many maturity levels should be set? Whether the score intervals between levels should be consistent? How to allocate them if they are inconsistent? These questions can be answered only when preset maturity levels and the corresponding relationship between score intervals and maturity levels are further verified and corrected. Verification and correction of mentioned contents were also realized through expert discussion, so for convenience, they are discussed together in the verification section. Also, in order to clarify the strengths and weaknesses of enterprises in assessment dimensions, the radar chart of dimensions was designed as the specific presentation of assessment results.

## 4. Verification of the Assessment System

Until now, determined contents were mainly obtained by theoretical analysis, so crucial attributes of this assessment system have not been verified through an application. Wernicke et al. [70] developed a framework for assessing the digital maturity of construction site operations. To examine and verify the framework, they conducted a case study on one construction site. The digital maturity of that site was firstly calculated by the proposed framework. Then they discussed, with the assessor, the detailed status of the digitalization of that site as well as the strengths and weaknesses. The consistency between the assessment results and the discussion results verified the proposed framework. Furthermore, the usability and benefits of this framework were also discussed. In this research, we also used expert discussions to verify our proposed assessment system. However, rather than conducting just one discussion with one assessment target, we conducted two rounds of expert discussions with experts from several construction enterprises. The first round was conducted to verify the scoring table and preset the maturity levels. The second round was conducted to verify those preset maturity levels. The overall verification process is shown in Figure 4. More details were discussed in our expert discussions.

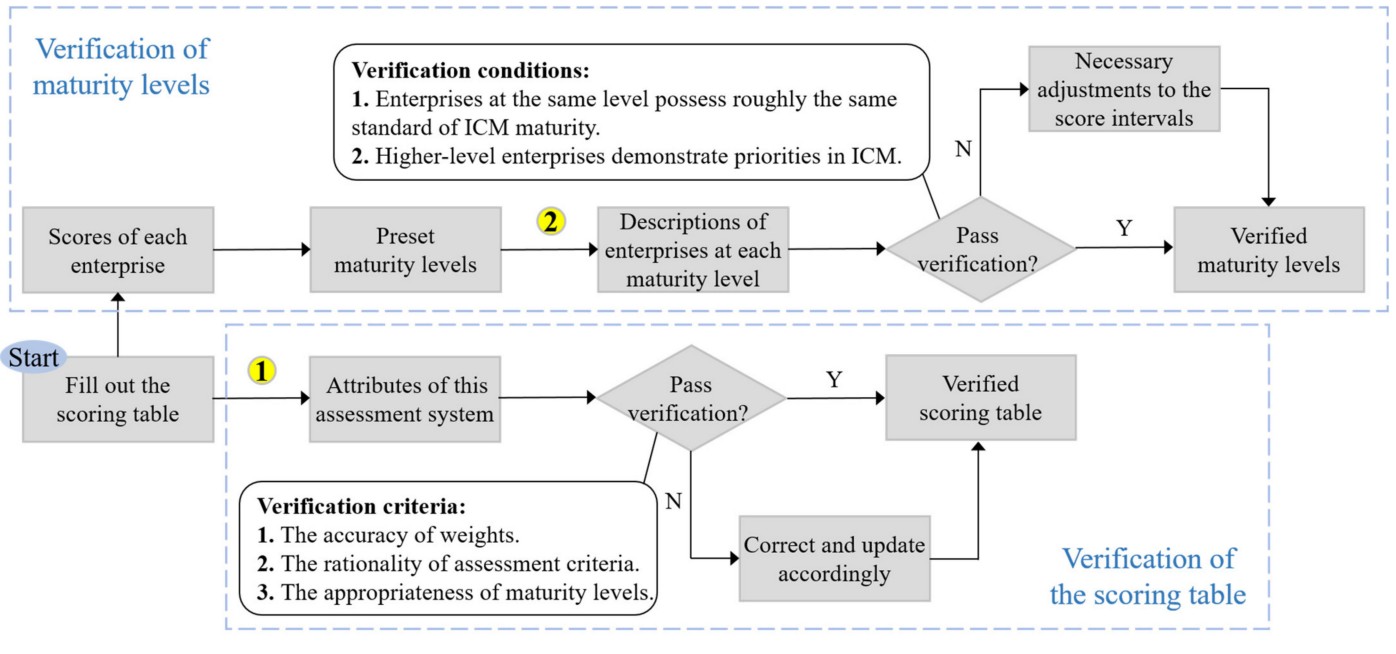

**Figure 4.** The overall verification process.

### 4.1. Verification of the Scoring Table

The scoring table was sent to leaders of many construction enterprises, and they filled out the scoring table according to the ICM maturities of their own enterprises. Except for collecting the scores of each enterprise, we discussed with the leaders the crucial attributes

of this assessment system. Through the feedback of these leaders, related contents of the scoring table were verified and updated.

### 4.2. Verification of Maturity Levels

After the collection of the scores of each enterprise, the distribution of all scores was obtained. Rough score intervals were divided by clustering all the scores, and thus the maturity levels were preset. According to the score intervals of the preset maturity levels, participating enterprises were differentiated into different maturity levels. Through detailed discussions with leaders from several representative enterprises at each maturity level, summary descriptions of the overall ICM maturity of enterprises at each maturity level were formed. The setting of the maturity levels must satisfy two criteria: (1) enterprises at the same level possessed roughly the same standard of ICM maturity; (2) enterprises at higher levels demonstrated relatively obvious priorities in ICM maturity compared with those at lower levels. Necessary adjustments to the set score intervals were conducted to satisfy the two principles, and the appropriate corresponding relationship between maturity levels and the scoring results was finally obtained. Thus far, all contents in the whole assessment system have been verified to ensure their accuracy, rationality and appropriateness.

## 5. Assessment System

### 5.1. Assessment Indicators and Dimensions

To obtain more extensive responses, we set two criteria for selecting the potential respondents for the questionnaire: (1) respondents with ample work experience in construction management or with ample knowledge of intelligent technologies; (2) respondents from as many professional fields as possible. Leaders from many different construction enterprises were interested in our study, so they helped us select employees from their enterprises according to these two criteria to answer the questionnaires. They told us that older employees tended to have more work experience while younger employees tended to have more knowledge about intelligent technologies. We offered them a QR code, which could be scanned to access our questionnaire and these leaders assigned qualified employees to complete the questionnaire. The questionnaires were collected two weeks after sending the QR code to leaders, and incomplete ones were deleted. Of the remaining questionnaires, 706 were considered valid. The basic information of the respondents is shown in Table 6. Respondents thought that the existing indicators were proper, so there was no need for eliminations or supplements. Furthermore, respondents provided us with suggestions on setting the assessment criteria, such as taking the operability of IT into account.

**Table 6.** Basic information of respondents.

| Professional Field | Amount | Professional Title | Amount | Working Post | Amount | Working Year | Amount |
|---|---|---|---|---|---|---|---|
| Roads and bridges | 483 | Primary | 131 | General supervisor | 166 | Within 5 | 73 |
| Tunnels | 60 | Middle | 337 | Specific supervisor | 346 | 5–10 | 140 |
| Traffic engineering | 84 | Vice-senior | 170 | Supervisor | 148 | 10–20 | 301 |
| Electromechanics | 10 | Senior | 68 | Enterprise administrator | 32 | More than 20 | 192 |
| Others [1] | 69 | | | Others [2] | 14 | | |

[1] Safety, structure, electric power, water transport, contract, experiment and engineering economy. [2] Vice-general supervisor, consultant and experimentalist.

There were twenty indicators in total, and their accurate weights were calculated using PCM. However, in order to make the data neat and easy to use, it was necessary to make adjustments within an appropriate range and keep the outcome as an integer. The scores of all assessment indicators and dimensions in the maturity scoring table were calculated by converting their weights, as shown in Table 7. They illustrated not only the present situation but also the future development trend of ICM.

(1) The first dimension had the highest weight, of nearly 40%, in the whole scoring table. It described the personnel organization, management and workflow of ICM. Effective

personnel organization and management are a crucial basis for every kind of enterprise and company to maintain competence, and it is the same for construction enterprises. Furthermore, as ICM is still rapidly developing, more and more suitable workflows will always be a key for construction management to develop more intelligence and for ITs and managerial approaches to maximize their superiorities. According to the expert discussion, nearly every enterprise has developed its own cloud platform. They work online, and their organizational framework was adjusted to adapt to the intelligent working mode.

(2) Indicators I-5, I-6, III-3, I-1 and V-2 have the top five highest weights. According to the expert discussion, they were all at present developing focuses for ICM. Most enterprises have been collaborating with researchers from institutes and universities to develop ITs and managerial approaches that these indicators describe. These ITs and managerial approaches have been realized to varying degrees among different enterprises. Due to the high weights these indicators possess, they are now decisive factors for a construction enterprise's ICM maturity.

(3) Indicators IV-3, V-3 and V-1 have relatively low weights. According to the expert discussion, only a few employees were contacted with the ITs and managerial approaches these indicators described. The development levels of these ITs and managerial approaches among different enterprises were pretty low. Publications about these ITs and managerial approaches were mostly limited to theoretical or prospective; they have not been comprehensively and maturely applied to ICM. However, as the publications imply, these ITs and managerial approaches have great benefit and considerable application potential to ICM, including VR [34], AR [35], prediction model [68], time-space conflicts management [67], etc. A number of construction enterprises have included these ITs and managerial approaches in their future development plans, and their weights will definitely increase in the future.

**Table 7.** Weights of dimensions and indicators.

| Assessment Dimension | Assessment Indicator |
|---|---|
| I. Organizational framework and working process (38) | 1. Working post setting (8), 2. Collaboration mode (7), 3. Personnel training (4), 4. Personnel assessment (1), 5. Workflow (9), 6. Transaction tracking (9) |
| II. Information collection and monitoring (12) | 1. Collection range (6), 2. Collection frequency (3), 3. Equipment integration (3) |
| III. Information transmission and aggregation (22) | 1. Transmission speed (6), 2. Information integration (7), 3. Information storage (9) |
| IV. Decision-making supported by visualization (12) | 1. Data visualization (2), 2. Knowledge base management (4), 3. Expanding reality (1), 4. Comprehensive decision (5) |
| V. Intelligent analysis and deduction (16) | 1. Auxiliary calculation (2), 2. Anomaly Identification (8), 3. Deduction and prediction (1), 4. Early warning and optimization (5) |

### 5.2. Maturity Scoring Table

The scoring table is attached in Appendix A. Assessment criteria for each indicator were set according to the application status of the relevant IT or managerial approach.

### 5.3. Maturity Levels

Maturity levels and corresponding score intervals were set according to the two criteria, as shown in Table 8. Five maturity levels were set, among which there was a particular level called "Minimum Intelligent Maturity". During the expert discussions, we found that ITs and managerial approaches described by many indicators have been well developed. Therefore, more than half of the scores in the scoring table were easily acquired by every construction enterprise. Enterprises with scores less than 60 must demonstrate

shortcomings in many aspects under this circumstance. Therefore, the lowest maturity level, "Minimum Intelligent Maturity", was set to conclude that these enterprises were not "intelligent" enough. As mentioned, many leaders of different construction enterprises filled out the scoring table and their scores were collected. The number of enterprises at level 2 was the largest. Few enterprises were located at the Minimum Intelligent Maturity level, and a few enterprises just entered level 3. There was not even one enterprise that entered level 4.

**Table 8.** Maturity levels and corresponding score intervals.

| Maturity Level | Score Interval |
|---|---|
| Minimum Intelligent Maturity | <60 |
| 1 | [60,70) |
| 2 | [70,80) |
| 3 | [80,90) |
| 4 | [90,100] |

*5.4. Radar Chart*

The radar chart was designed to compare the relative development of each dimension, but the weights of each dimension were different so that their full scores in the scoring table were also different. When using the radar chart, scores of each dimension in the scoring table should be converted into a centesimal system:

$$R_i = \frac{T_i}{W_i} \times 100$$

In this function, $R_i$ is the score of dimension $i$ in the radar chart, $T_i$ is the score of dimension $i$ in the scoring table and $W_i$ is the weight of dimension $i$ (shown in Table 7). Then the pentagon representing the ICM maturity of the enterprise in these dimensions was drawn.

**6. Case Study**

The proposed assessment system was examined in a case study of two construction enterprises, A and B. These two enterprises are both located in Hangzhou, a city in southern China. Enterprise A mainly focuses on the construction of highways and canals; enterprise B mainly focuses on the construction of highways and railways. One leader from each of the two enterprises filled out the maturity scoring table according to the actual situations of their enterprises.

*6.1. Assessment Results*

Their assessment results, using the maturity scoring table, were 78 and 81, respectively. Therefore, the ICM maturities of enterprises A and B were very close. They both almost entirely satisfied the demands of level 2, and enterprise B had just entered level 3. Furthermore, to show their strengths and weaknesses in each dimension, a radar chart of their ICM maturities was drawn, as shown in Figure 5.

Then two discussions with the two leaders were conducted for detailed situations of their ICM maturities, as shown in Table 9. Enterprise A and B paid great attention to the development of ICM, and they made specific development plans for the introduction and application of several ITs and managerial approaches. However, their unsystematic development plans caused different kinds of deficiencies in each dimension and led to relatively elementary maturity levels thereafter. Enterprise A and B are located in the same city, and they have known each other for years. They both commented that their development situations in ICM are very close, and they admitted that their maturity levels are relatively elementary. Not enough new roles have been set in enterprise A, and existing roles have not been thoroughly adapted to ICM, leading to a burden on the

newly developed workflow. It is the main reason, according to our assessment system, for enterprise A to have a slight gap in scores compared with enterprise B.

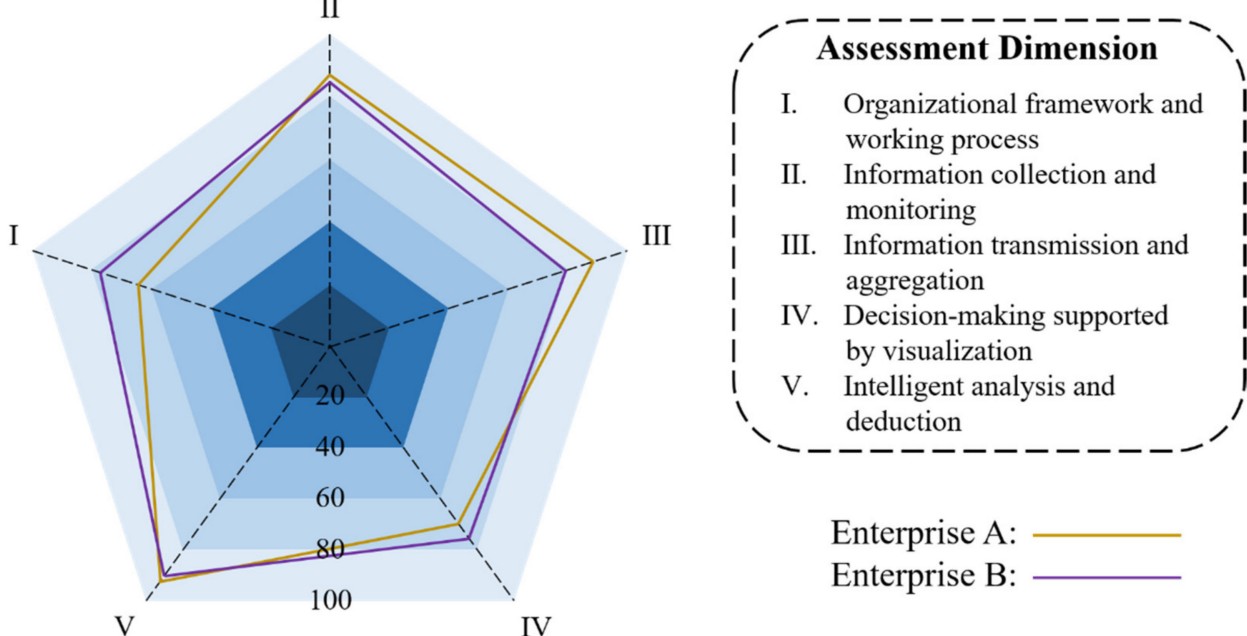

**Figure 5.** The radar chart of enterprises A and B.

**Table 9.** Detailed situations of the ICM maturities of enterprises A and B.

| Assessment Dimension | Strength | Weakness |
|---|---|---|
| I. Organizational framework and working process | A and B: Develop a construction management platform, and transactions are strictly tracked.<br><br>A: Monthly personnel assessment.<br>B: Many new roles are set. | A and B: Insufficient personnel training. Some employees are still not skilled with the platform, and workflows are still not clear.<br><br>A: Only a few new roles are set. |
| II. Information collection and monitoring | A and B: Many data collection devices are arranged on construction site.<br>A: Enough mobile phones for management.<br>B: A few attempts at equipment integration. | A: Almost no equipment integration, efficiency and accuracy of data collection are low. |
| III. Information transmission and aggregation | A and B: All data and information collected are stored.<br>A: Efficient information aggregation (a new role was set for this). | A and B: Inadequate coverage of network signal on-site; real-time upload and receipt cannot be guaranteed. |
| IV. Decision-making supported by visualization | A and B: Many kinds of important data and information are displayed in real time by adequate display devices. | A and B: Poor interaction between different kinds of models, no application of VR, AR or MR. |
| V. Intelligent analysis and deduction | A and B: Many intelligent functions and algorithms are developed.<br>A: Employed a software system team. | A and B: The frequency of use is unstable. |

### 6.2. Validation of the Assessment System

Despite many existing weaknesses and deficiencies for these two construction enterprises, the biggest problem for them is the low capacities of organizational management, which are caused by the incompletely adjusted working posts. Even if there are ITs and managerial approaches developed, they cannot be efficiently applied. Therefore, we strongly recommend these two enterprises set more new roles first, and then other measures can be applied, such as training employees more often, improving the network signal on site, introducing VR, AR or MR, etc. After the discussions, these two leaders claimed that they would adjust their development plans according to our recommendations.

In total, these two leaders spoke highly of the usability of the assessment system. They commented that the assessment criteria helped them to reach deeper into their actual development situation with ICM, and the assessment dimensions are essential for them to find their deficiencies and weaknesses in detail. Therefore, the case study proved the validation of the proposed assessment system, and it can provide not only overall and specific representations of the ICM maturity of a certain enterprise, but also targeted development plans thereafter.

### 6.3. Discussion

There should also be a potential, anticipated or typical development path to the desired target state after evaluation [81]. The ultimate purpose of the maturity assessment of the ICM is to efficiently and accurately improve its maturity. After assessment using the maturity scoring table, construction enterprises can further understand details of their weaknesses through the indicators with lower scores, and these are the aspects that need to improve most. To provide specific improvement plans for construction enterprises, this study discussed improvement strategies from two perspectives.

The first perspective was the detailed ITs and managerial approaches. This study extracted and summarized ITs and managerial approaches that could improve the maturity of ICM during the extraction of the assessment indicators. They could be used to help construction enterprises that have been assessed to discover and fill the present gaps. These ITs and managerial approaches are listed in Table 1, where they are very detailed and specific. They were simply classified according to their effects on ICM, and their sources are contained in the table, which can be searched and consulted by enterprises to understand the development and application methods of any IT or managerial approach in detail.

The second perspective was the framework of enterprises at level 4. Organizations engaged in information, digitalization, or intelligence have similar frameworks, which include many layers. However, each layer in the framework of different kinds of organizations consisted of different components. The frameworks of highly developed organizations were powerful and comprehensive, where all layers had undergone extensive development. When a certain construction enterprise had the highest maturity of ICM, it must have the fourth maturity level and developed almost every IT and managerial approach in Table 1. Under this circumstance, each layer in its framework possessed adequate components. Representative components are summarized according to the sources of ITs and managerial approaches in Table 1, as shown in Figure 6. This framework can be used for reference by construction enterprises to supplement their own framework and finally continuously improve their ICM maturities.

In general, construction enterprises are able to obtain suitable improvement plans synthetically from these two perspectives after assessment, as shown in Figure 7.

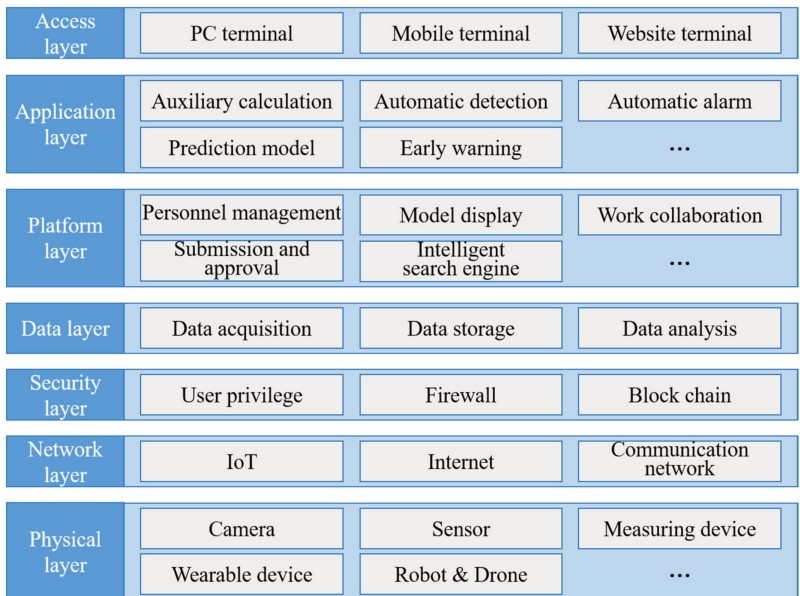

**Figure 6.** Framework of enterprises at level 4.

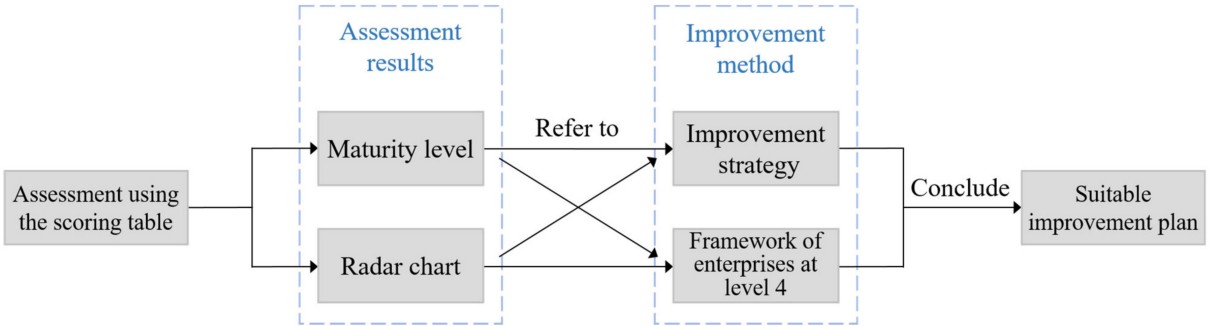

**Figure 7.** Method to obtain suitable improvement plans.

## 7. Conclusions

ICM is rapidly developing at present, but there is still a lack of systems, methods or even indicators to systematically assess the maturity of ICM. In this study, we developed a maturity assessment system through literature reviews, questionnaires, expert discussions and a case study. The maturity assessment system consisted of a maturity scoring table, maturity levels and a radar chart of dimensions, which can be used by construction enterprises to assess their ICM maturities and formulate suitable future improvement plans. First, fill out the maturity scoring table based on the ICM development situation of a certain construction enterprise. Then the score can be converted into a maturity level and a dimension radar chart. The position of the enterprise in the whole construction industry and its own strengths and weaknesses can be accurately understood. Finally, a suitable improvement plan for this construction enterprise can be created with reference to the improvement strategy and the framework of enterprises at level 4.

The maturity scoring table consists of five assessment dimensions and twenty assessment indicators. When using it, assessors need to score each indicator based on their subjective judgment of their own construction enterprises. Since the subjective difference is inevitable, it is strongly recommended that the ICM maturities of each enterprise are assessed by more than one leader, and the average of their scores is taken as the final result. According to the scoring table, developing ITs and managerial approaches, which support organizational framework and working process, are of great significance for construction enterprises to reach high ICM maturity. Many enterprises do not pay enough attention to advanced managerial approaches because they have not realized the unimagined progress

that these approaches can bring to them. During our discussion with leaders in construction enterprises, we found that many enterprises had already developed adequate IT and managerial approaches. Unfortunately, a large number of ITs and managerial approaches remained unused or insufficiently used for a lack of suitable organizational frameworks and working processes, leading to a low ICM maturity with a huge amount of resource waste.

The study has two limitations. First, the study used the maturity scoring table as the assessment method because of the high complexity of the assessment process of ICM maturity. Although the scoring table is accurate and reliable, it is not efficient enough. Future research can use the assessment indicators in this study to establish a more efficient maturity assessment methods for ICM, such as the assessment methods consisting of yes or no questions, flowcharts or single-choice questions. Second, the assessment indicators and their weights in this study represented the developing situation of ICM in the present and short-term future. There is still a long way for construction management to thoroughly reach intelligence since most construction enterprises are located at level 2, and there is nearly none that have entered level 4. However, there will be a time when most enterprises have entered level 4 because of the efforts that the whole construction industry is making. Meanwhile, more ITs and managerial approaches will come into use and serve as indicators. The weights of all indicators must change with time. Future studies are recommended to add new indicators and correspondingly adjust the weights of all indicators. For example, the ITs and managerial approaches toward automation are rapidly developing and will occupy more and more weight [82,83].

**Author Contributions:** Conceptualization, J.-R.L.; methodology, C.L. and J.-R.L.; validation, Z.-Z.H. and Y.-C.D.; formal analysis, C.L., C.Y. and W.Z.; investigation, C.L. and J.-R.L.; resources, C.Y. and W.Z.; data curation, C.L.; writing—original draft preparation, C.L. and J.-R.L.; writing—review and editing, C.L., J.-R.L. and Y.-C.D.; visualization, C.L.; supervision, Z.-Z.H., J.-R.L. and Y.-C.D.; project administration, Z.-Z.H., J.-R.L. and Y.-C.D.; funding acquisition, J.-R.L. and Z.-Z.H. All authors have read and agreed to the published version of the manuscript.

**Funding:** This research was funded by the National Key R&D Program of China (grant number 2018YFD1100900) and the Science and Technology Plan Project of the Zhejiang Provincial Department of Transportation (grant number 2020061).

**Data Availability Statement:** Data sharing is not applicable.

**Conflicts of Interest:** The authors declare no conflict of interest.

## Appendix A

**Table A1.** The maturity scoring table for ICM.

| Dimension | Indicator | Criterion | Score |
|---|---|---|---|
| I. Organizational framework and working process | 1. Working post setting: Adjust former posts and set specific responsibilities and corresponding organizational relationships for all posts. | Reasonable, perfect personnel allocation and gross wage. | 8 |
| | 2. Collaboration mode: Conduct online personnel management and workflow interaction based on the management platform. | Real-time uploads, reminders and feedback. | 7 |
| | 3. Personnel training: Train personnel in a variety of ways to adapt to the working mode of ICM. | Check regularly and trace the training data. | 4 |
| | 4. Personnel assessment: Use a variety of data sources to assess the attendance and performance of personnel. | Quantitative, qualitative and objective assessments. | 1 |
| | 5. Workflow: Assign designated, responsible personnel to complete the workflow of each task with a clear work sequence. | Smooth workflow with high efficiency. | 9 |
| | 6. Transaction tracking: Record and track the processing flow and relevant responsible personnel for all transactions. | Clearly record the process and responsible person. | 9 |

**Table A1.** *Cont.*

| Dimension | Indicator | Criterion | Score |
|---|---|---|---|
| II. Information collection and monitoring | 1. Collection range: Collect as many types of data and information as possible on-site and make the collection range as wide as possible. | Collect comprehensively and all key areas covered. | 6 |
| | 2. Collection frequency: Collect data and information as frequently as possible to improve their continuity. | Avoid interruptions in data and information acquisition. | 3 |
| | 3. Equipment integration: Develop equipment that collects multiple data and information simultaneously and efficiently. | Improve the accuracy of data and information collection. | 3 |
| III. Information transmission and aggregation | 1. Transmission speed: Improve the transmission speed to ensure the timeliness of data and information transmission. | Ensure real-time uploading and receiving on-site. | 6 |
| | 2. Information integration: Integrate, fuse, summarize and transform the collected data and information. | Automatic preprocessing of data and information. | 7 |
| | 3. Information storage: Archive and save the collected data and information to support efficient utilization and security management. | Store all data for the whole life cycle of the project. | 9 |
| IV. Decision-making supported by visualization | 1. Data visualization: Model, visualize and simulate using all kinds of construction data and information. | Concrete, intuitive and accurate visualizations. | 2 |
| | 2. Knowledge base management: Upload construction data and information to the platform and set search functions for viewing. | Comprehensive categories and accurate search results. | 4 |
| | 3. Expanding reality: Assist scheduling and management with the help of VR, AR, MR and other extended reality technologies. | Widely used in the complete workflow. | 1 |
| | 4. Comprehensive decision: Real-time display of the data and information being monitored and collected. | Display on a variety of devices widely. | 5 |
| V. Intelligent analysis and deduction | 1. Auxiliary calculation: Intelligently calculate the schedule, cost, etc., with collected data and information. | Introduce intelligent computing for all calculation processes. | 2 |
| | 2. Anomaly identification: Identify occurring abnormal conditions, including automatic detection of construction results and automatic alarm of unsafe behaviors, etc. | Comprehensive categories, fast detection and identification speed with high accuracy. | 8 |
| | 3. Deduction and prediction: Establish a prediction model based on collected data and information to predict the work focus and potential risks in the next stage. | The model considers all types of data and information, real-time update. | 1 |
| | 4. Early warning and optimization: Adjust the management plan according to the prediction results to avoid possible risks and improve the project management ability. | Real-time optimize the management plan and implement it on time. | 5 |

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
