# Peer review of "Maturity Assessment of Intelligent Construction Management"

_buildings, doi:10.3390/buildings12101742_

Round 1

Reviewer 1 Report

The paper deals with the problem of proper construction management, which has a decisive influence on the reduction of potential risks when implementing investment and construction projects and making necessary conditions for timely and high-quality completion of projects within the planned budget.

The application of a large number of intelligent information technologies and advanced managerial approaches such as management platform, visualization, workflow, production, quality control, schedule, etc. have brought on rapid development to Intelligent Construction Management. In the Introduction, the authors point out the lack of effective methods to assess the maturity of ICM. Accordingly, the main focus of the authors was to develop a maturity assessment system for intelligent construction management, which is of great significance to improve construction enterprises and discover their potential problems. It was formulated through literature reviews, questionnaires, expert discussions, and a case study.

A maturity scoring table containing five assessment dimensions (among others: The organizational framework and working process, the collection and monitoring of information, the decision making supported by visualization) and 20 assessment indicators were developed, and the corresponding maturity levels and the dimensions radar chart were designed to visualize the strengths and weaknesses in each dimension.

A case study of the assessments of two construction enterprises A and B was conducted to validate the proposed assessment system that can be used by construction enterprises to quantitatively assess their ICM maturities and get both general and specific assessment results.

These two enterprises were located in China in Hangzhou and related, respectively, to the construction of highways and canals and the construction of highways and railways.

As a result, the assessment system presented by the authors can be used in construction enterprises to obtain vivid assessment results and improvement plans. What is very positive, the authors themselves present the limitations of the proposed analysis, which uses not efficient enough the maturity scoring table as the assessment method. The authors recommend future research by adding new indicators and correspondingly adjusting the weights of all indicators.

In conclusion, I consider the paper to be positive and recommend it for publication in the Buildings Journal.

Author Response

Many thanks for the reviewer’s affirmation and recommendation. Your brief rehearsal exactly tells what we wanted to research for and your conclusion to our paper covers precisely every point of our research focus. Thank you very much again.

Reviewer 2 Report

The paper's contribution mainly covers developing a system including a score table and radar chart to assess the maturity level of Intelligent Construction Management (ICM) for construction institutes. The authors did a good effort in evaluating the system with many methods. I believe the study has value to encourage companies in adapting new technologies. I have only one notice, the paper is lengthy and can be reduced to make it easy to read. Much repeated information I noticed on the paper.

Author Response

Thanks for the comments and suggestions. We have shortened the paper by deleting repeated information. In all, the paper was shortened from 23 pages to 22 pages. Before shortening, we wrote some contents in the beginnings of each sections to make abstracts, but these contents will be repeated again in detail later. We have deleted these “abstracts” in the beginning of each section to make the whole paper shorter and neater.

The contents in the following sections are shortened: section 3 (subsection 3.1, 3.2, and 3.3), section 4, section 5 and subsection 5.2.

Reviewer 3 Report

This paper presents the authors’ work on developing a system for assessing enterprises’ level of maturity of Intelligent Construction Management (ICM), which includes five levels of maturity and can help enterprise to develop improvement plans. To achieve the research objective, five tasks were carried out, including (a) identifying assessment indicators via literature review and expert discussion, (2) designing questionnaire to collect industry experts’ opinions on the indicators and their importance, (3) calculating the weights of indicators using Precedence Chart Method (PCM), (4) creating the maturity scoring table, and (5) validating the system by using questionnaire and case study. Overall, this paper investigated a key topic in intelligent construction management, and the study is presented with a good structure and clear motivation. However, the following aspects should be clarified or improved.

1 Methodology

1) This study used literature review and expert discussion to identify the assessment indicators, but it is not clear in the paper which indicators were from literature/expert discussion, and what criteria were used for determining maturity level (minimum level and level 1-4).

2) Questionnaire is a key part of the method, as it was used to develop and validate the proposed framework, but many aspects of the questionnaire are not clear yet. The following aspects should be clarified: (1) the criteria for selecting the potential participants for the questionnaire (who are eligible to answer the questions, considering e.g., their position and work experience), (2) the number of questionnaires distributed and the response rate, and (3) the participants’ opinion on the indicators. These aspects are important for assessing this study’s method and justifying the proposed framework.

3) Please consider combining step 2 and step 3 of the method, they are both about determining the weight of indicators.

2 Writing quality

1) The writing quality of sections after the literature review is not as good as the previous sections. Errors in grammar and obscure meaning of text are noticed.

a) Line 81 – ‘it need’

b) ‘line 88, 155, 170’ – ‘research’ is uncountable, please use other words, such as ‘studies’

c) Line 181 – ‘can be the established by extracting …’

d) ‘Line 274’: not only … but also…

e) Line 610: ‘develop ITs and managerial approaches… is of great significance’, develop -> developing

f) Line 493: ‘ITs and managerial approaches … have been developed to different maturities so more than half of the scores are easily acquired by every construction enterprise’, this sentence has obscure meaning.

g) Line 554: ‘these two leaders spoke high of the usability of the assessment system.’ High -> highly.

2) The conclusion section should be improved. Briefly introduce the problem solved in this study and the methodology, interpret the result, and discuss its impact.

3 Suggestion

It is suggested that the authors proofread the manuscript and address the questions mentioned above.

Author Response

Dear reviewer, thank you very much for your kind comments and suggestions, we have prepared the point-by-point response, please check the attached file for detail.

Round 2

Reviewer 3 Report

All my concerns/questions have been properly addressed, I have no more questions about this paper.